# Surveillance of Colorectal Cancer (CRC) in Cystic Fibrosis (CF) Patients

**Fabio Ingravalle** [1,*], **Giovanni Casella** [2], **Adriana Ingravalle** [3], **Claudio Monti** [4], **Federica De Salvatore** [5], **Domenico Stillitano** [6] **and Vincenzo Villanacci** [5]

1  School of Specialization of Hygiene and Preventive Medicine, University of "Tor Vergata" Rome, 00161 Rome, Italy
2  Gastroenterologist, General Practitioner Limbiate, ATS Lecco-Brianza, 20812 Monza Brianza, Italy; caselgio@tiscali.it
3  School of Specialization in Anesthesia and Critical Care Medicine, Sapienza University of Rome, 00161 Rome, Italy; adriana.ingravalle@gmail.com
4  General Practitioner Barlassina, ATS Lecco-Brianza, 20825 Monza Brianza, Italy; c.monti_2012@libero.it
5  Institute of Pathology, ASST Spedali Civili Brescia, 25126 Brescia, Italy; federica.desalvatore@libero.it (F.D.S.); villanac@alice.it (V.V.)
6  Endoscopy Unit, Desio Hospital, ASST-Monza, 20811 Monza Brianza, Italy; domstilly@tiscali.it
*  Correspondence: fabio.ingravalle@gmail.com; Tel.: +32-045-035-08

**Abstract:** Cystic Fibrosis (CF) is the commonest inherited genetic disorder in Caucasians due to a mutation in the gene CFTR (Cystic Fibrosis Transmembrane Conductance Regulator), and it should be considered as an Inherited Colorectal Cancer (CRC) Syndrome. In the United States, physicians of CF Foundation established the "Developing Innovative Gastroenterology Speciality Training Program" to increase the research on CF in gastrointestinal and hepatobiliary diseases. The risk to develop a CRC is 5–10 times higher in CF patients than in the general population and even greater in CF patients receiving immunosuppressive therapy due to organ transplantation (30-fold increased risk relative to the general population). Colonoscopy should be considered the best screening for CRC in CF patients. The screening colonoscopy should be started at the age of 40 in CF patients and, if negative, a new colonoscopy should be performed every 5 years and every 3 years if adenomas are detected. For transplanted CF patients, the screening colonoscopy could be started at the age of 35, in transplanted patients at the age of 30 and, if before, at the age of 30. CF transplanted patients, between the age of 35 and 55, must repeat colonoscopy every 3 years. Our review draws attention towards the clinically relevant development of CRC in CF patients, and it may pave the way for further screenings and studies.

**Keywords:** cystic fibrosis; colorectal cancer; colonoscopy screening; colonic polyps; screening strategies

## 1. Introduction

Cystic Fibrosis (CF) is one of the most common inherited genetic disorder in Caucasians. A mutation in Cystic Fibrosis Transmembrane Conductance Regulator (CTFR) gene is the cause of the disease [1]. The CTFR gene encodes for the chloride ion channel, belonging to adenosine triphosphate's family. This families of channels transport chloride ions into and out of secretory epithelia cells, and they are regulated by protein kinase A/cyclic adenosine monophosphate (cAMP) [2]. CFTR protein is widely expressed also in the mucosa of the gastrointestinal tract [3], and its dysfunction can lead to several Gastrointestinal disorders, such as Distal Intestinal Obstruction Syndrome (DIOS), Constipation, Rectal Prolapse, Cholelithiasis, Gastro-Esophageal Reflux Disease, Exocrine Pancreatic Insufficiency (EPI) and GI Cancer as Colo Rectal Cancer (CRC) and Pancreatic Cancer [4]; CFTR was identified as the gene causing CF in 1989 [5].

After the gene discovery, more than 2000 mutations of CFTR gene have been identified, but the functional importance is known only for 300 of them [6]. The most common CFTR mutations may be divided in 6 classes (Figure 1): (1) in class 1 mutation group, the CFTR protein synthesis is severely reduced or absent; (2) in class 2 mutation group, the CFTR protein is misfolded or early degraded; (3) class 3 mutations impair the regulation of the CFTR channel; (4) class 4 mutations impair the conductance of CFTR channel; (5) in class 5 mutation group, there is an alteration of the promoter or in splicing process; (6) in class 6 mutation, there is an accelerated turnover and reduced apical expression of CFTR channel [7,8] Cystic Fibrosis (CF) is a life-threatening autosomal recessive disease and estimates predict that it affects 30,000 individuals in United States and 70,000 individuals worldwide [9].

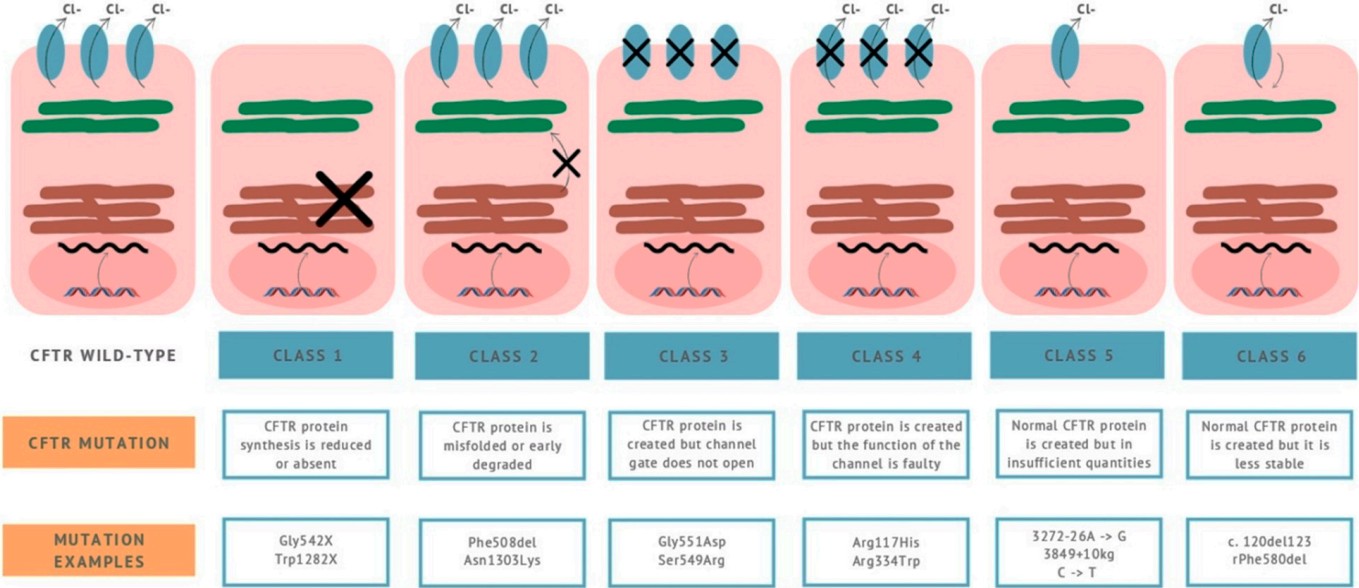

**Figure 1.** CFTR Mutation classes.

CF is considered as a multisystem disease that affects principally respiratory system, where lungs are the most involved organs [10]. The progressive improvement in early diagnosis and in managing respiratory manifestations has allowed to increase the survival rate and life expectancy in CF patients. In particular, life expectancy is improved from 30 years in 1986 to 46.2 years in 2011. CF should be always treated by a multidisciplinary team dedicated to CF patients care [11].

In the United States, physicians of Cystic Fibrosis Foundation stressed the involvement in CF of gastrointestinal tract and hepatobiliary ducts. They established the "Developing Innovative Gastroenterology Specialty Training Program" to increase the research on this field [12].

## 2. Cystic Fibrosis and Colon Cancer

Cystic Fibrosis (CF) should be considered as an inherited colorectal cancer (CRC) syndrome [13]. The risk to develop a CRC is 5–10 times higher in CF patients than in the general population [14] and ("even greater in CF patients on immunosuppressive medication after organ transplantation"). The relative risk is increased 30-fold respect to general population [15] and even greater in CF patients on immunosuppressive medication after organ transplantation.

An early onset of colon adenoma and an increased risk of colorectal cancer (CRC) in adults with CF has been reported by different centers in United States [16], Australia [17] and Canada [18].

## 3. Epidemiology

The largest study which has assessed the risk of cancer in CF patients included 28.511 patients in United States and Canada and, approximately, 24,500 CF patients across 17 European countries between 1985 and 1992 [4]. In both geographical areas, the overall risk of cancer was similar to that of the general population, but the risk of digestive tract cancer was significantly increased (Odds Ratio OR −6.5). The increased risk was most dramatic in CF patients aged 20–29 years (OR > 20) and for cancer affecting the Pancreas (OR 31.5), Esophagus (OR 14.3) and Bowel (OR 9.3)

Another large study performed by the same team [19] in 28.858 CF patients in North America between 1990 and 1999 obtained similar results.

An additional increase in the incidence of digestive tract tumors has been observed in patients undergone transplant procedures, probably related to sustained and intense immunosuppressive therapy. In the large study of Maissoneuve et al. [19] of about 28.858 patients, including 1068 post transplantation, the standardized incidence ratio (SIR) for all cancers in post transplantation patients was 6.3 and for digestive tract tumors was 21.2 [19]. Meyer et al. [20] reported that, in 1 North American Transplant Center, 4/65 (6.2%) of CF lung transplanted patients surviving beyond 6 months developed fatal colonic adenocarcinoma [20]. Furthermore, 7/20 (35%) of these patients subjected to screening colonoscopy showed polyps ranging from 2 to 30 mm in diameter [20].

## 4. Pathophysiology

The mechanisms for increased risk of CRC in CF are unclear [21]. CFTR gene has also a tumor suppressor function, and CFTR mutation can be associated to the pathogenetic pathway in intestinal tumor formation [22]. CFTR plays a "key role" in secretion of chloride and bicarbonate ions in the colonic mucosa [21]; the transport of ions helps to check the movement of water across intestinal barrier, and its function is fundamental to produce intestinal mucus. The mucus protects epithelial layer and avoids dehydration of the epithelial layer. Hence, a dehydration of the intestinal mucosa may lead to a bacterial overgrowth [23].

The absence of CFTR is associated with dysregulation of immune response, intestinal stem cells and growth signaling regulators [24]. Important cancer risk factors in CF population as high fat diet, (cancel low physical activity) and diabetes mellitus are more evident than in general population [25]. Furthermore, the increasing use of antibiotics in CF population may result in intestinal stasis, chronic gut inflammation and malabsorption of nutrients. These events can increase the risk of colonic adenoma or a CRC [26]. Moreover, intestinal microbial dysbiosis plays an important role in tumorigenesis, as better explained in the following chapter. Dysbiosis can indirectly damage the DNA, activate the oncogenic pathways, induce the production of mutagenic metabolites and suppress antitumor immunity [27]. CF carriers have 50% of CFTR anion channel activity than controls [28] determining a reduction of anion transport in epithelial cells. CFTR carriers are at greater risk for CF related conditions, and this condition may help clinicians and patients to increase screening and prevention approaches [29].

Biallelic inactivating germline mutations in the CFTR gene found on chromosome 7 determine Cystic Fibrosis, the most common autosomal recessive disease among people of European (primarily Northern Europe) ancestry [30]. CFTR produces an mRNA transcript of 6128 nucleotides [31] encoding a protein of 1480 amino acids that works as a Chloride (CL) and Bicarbonate ($HCO_3$) anion channel. CFTR is located on the apical surfaces of luminal epithelia [32].

Chloride channel, including CFTR, is important in homeostasis of the gastrointestinal (GI) tract for several important functions as: osmoregulation, transport of major ions across epithelia, (polarity of cells—cancel), glucose metabolism, (cellular autophagy—cancel) and (protein turnover—cancel), (migration of cells—cancel), mucus secretion, innate and adaptative immune responses, (cell–cell interaction—cancel), (membrane potential—cancel), mitochondrial function and related oxidative stress, tissue inflammation, microbiota com-

position, cellular pH, programmed cell deaths and intestinal stem cell regulation [33]. GI tract is constantly exposed to environmental insults, and the dysregulation of these ion channels may contribute to carcinogenesis [33].

CTFR Is expressed in the whole intestinal tract with a gradient of decreasing expression from duodenum to distal ileum [34]. In the colon, CFTR expression is greater in cecum and proximal colon than in distal colon [32]. In small and large bowel, CFTR expression is strongest at the base of the crypt, typical location of the intestinal stem cell compartment [35].

CFTR determines water homeostasis in intestinal tract [36] due to membrane-spanning domains of CFTR that permit the production of an aqueous channel for the passage of Cl and HCO3 ions from the cytoplasm of epithelial cells to the intestinal lumen.

CFTR is also involved in Sphingosine-1 Phosphate (S1P) extracellular transport playing a critical role in the regulation of inflammatory signaling and cell adhesion [37].

CFTR contains a Cytoplasmic C terminal PDZ (post synaptic density protein 95 (PSD-95)-Drosophila discs large tumor suppressor (DLG1)- zona occludins 1 (ZO-1)- binding motif that interacts with PDZ-containing proteins that regulate intracellular signaling and the actin cytoskeleton [38].

In conclusion, CFTR function is fundamental for ion and water homeostasis in GI tract. CFTR localization to the intestinal crypt stem cell compartment permits to influence stem cell function, and they, at the same time, may be considered cancer progenitor cells [39].

## 5. Screening

Only limited data are available on systematic colonoscopy screening in CF patients for intestinal cancers [40]. The limited data suggest that the most part of colonic neoplasms are localized in right colon [20]. In CF patients, there are other important factors that can increase the risk of CRC, such as family history of CRC, sex, age of CF diagnosis, presence or absence of meconium ileus, diabetes, distal intestinal obstruction syndrome and pancreatic insufficiency [21].

Individuals with a family history of Colon Rectal Cancer (CRC) should perform a colonoscopy every 5–10 years starting at age 40 years [41]. Individual with a Lynch Syndrome should undergo colonoscopy every 1–2 years starting at age 20–25 years [42]. Screening in CF patients falls between the risk of these different groups, but it is important to underline that the risk in lung transplanted CF patients is 30-fold increase compared to general population and it **is** higher than Lynch Syndrome patients [43]. Although, it is important to consider that CF patients have an up to 30-fold increased CRC risk compared to average US individuals and CRC deaths predicted among them were less than reported for the US general population (19.1 and 22.3 versus 27.8 per 1000) due to their more elevated other cause mortality (70% of deaths in CF individuals are related to cardiorespiratory causes) [44].

## 6. Dysbiosis in Cystic Fibrosis (CFTR Deficiency) Patients

Bacterial dysbiosis is commonly observed in the gut of CF patients [45]. Loss of CFTR causes a dysregulated environment in the intestinal lumen that is associated with disruption of the mucus layer and exacerbated by some factors present in CF patients as nutrient malabsorption, high fat diets and antibiotic therapy. The degree of dysbiosis is associated with the severity of CF GI diseases, GI inflammation and nutrient uptake deficiency [45]. In CF patients, a reduction of protective bacteria as Acinetobacter Iwoffii and Lactobacilliales members was noted and associated to a rise in Mycobacteria species and Bacteriodes Fragilis determining an increase of infection and altered immunomodulation in GI tract [46]. Meeker et al. [47], in their experiments with CFTRKO mice, reported that CFTR mutation alters the fecal GI microbiome with an increase >250-fold of Escherichia/Shigella and a depletion of Lachnoclostridium and Parabacteroides. CF patients showed decreased microbial diversity with a decreased abundance of Butyrate producing bacteria such as Ruminococcaceae and Butyricimonas associated to an increase of Taxa Bacteria as Actinobacteria and

Clostridium that they determine development and progression of CRC [48]. Dysbiosis determines a dysregulation of physical barrier integrity with illicit bacterial contact with the intestinal epithelia determining an activation of innate and adaptative immune system fundamental to the onset of direct and indirect pro-inflammatory response [49].

### 7. CF Patients Are at High Risk for Colorectal Cancer (CRC)

The expected average life expectancy of newborn is increased in the last years with a median age of 44 years thanks to the development of more effective therapeutic modalities for pulmonary dysfunction [50]. In CF patients a longer survival determines an increased risk for developing specific cancers. Maissonevue et al. [14], in a 20-year epidemiological study using data from the US CF Registry, reported that in these CF patients the risk of CRC was increased 6-fold. Endoscopic screening studies evidenced that CF patients showed larger, more aggressive colonic polyps than non-CF individuals [13]. These studies reported that 50% of CF patients by 40 age developed adenomas; moreover, half of these tumors were classified as aggressive advanced adenomas [16]. 3% of CF patients by 40 age developed adenocarcinoma [40]. The Cystic Fibrosis Foundation has declared CF as a hereditary colon cancer syndrome [21]. CRC endoscopic screening should be started in CF patients by age forty and in immunocompromised CF lung transplant patients, at particular risk for cancer, by age thirty [21].

### 8. CFTR Carriers Are also Susceptible to Gastrointestinal (GI) Cancers

Miller et al. [29], in a population-based study comparing the prevalence of CF related conditions in more than 19,000 CFTR heterozygous mutant carriers and 99,000 healthy controls, noted an increased prevalence of GI cancers (colon, stomach and other GI organs) in 44% of CF carriers and 96% of these were under the age of 46 years [29]. In United States (USA), 3–4% of all population (>10 million people) are heterozygote carriers for inactivating CFTR mutations [51]. In USA, the average age for sporadic Colon Rectal Cancer (CRC) is 68 years in males and 72 years in females with a similar average age for gastric cancer.

### 9. Role of Fecal Immunochemical Test (FIT) and Other Fecal Tests

The Task Force of Cystic Fibrosis [21] does not recommend fecal immunochemical test (FIT) (on whole CF patients, but on other hand results—cancel), however there are some studies that suggest that screening of CRC by FIT may be considered more cost effective in CF individuals than colonoscopy screening [21]; another analysis suggests that this strategy can be more effective (46 vs. 44 LYG) and less costly (2.5 vs. 2.6 million of dollars) than colonoscopy [52]. (Nevertheless, the validity of FIT has yet to be confirmed in clinical studies, before using FIT screening in CF patients [21]—cancel). However, at this moment, no study combines and compares the predictivity of FIT screening versus colonoscopy screening for CRC [4]. According to international literature, FIT may be suitable for the general population but not indicated, at this moment, for CF patients [21]. Multi-targeted stool DNA testing should be considered, but it has never been evaluated in CF patients and any positive test result requires a confirmatory colonoscopy [21] that should be always considered as "the gold standard tool" to detect and remove colonic polyps.

### 10. Role of Computed Tomography Colonography (CTC)

Computed Tomography Colonography (CTC) cannot be recommended to diagnose Colonic Neoplasms in CF patients [21]. CTC does not permit colonic polypectomy, and it may be considered a source of mistake in the evaluation of small lesions [21].

### 11. Role of Colonoscopy in CF Patients

Cystic Fibrosis Task Force [21] recommends colonoscopy screening in CF patients from the age of 40 (Figure 2). In case of negative results, it suggests repeating colonoscopy every 5 years [14] and every 3 years if adenomas are detected [4]. CF is associated with earlier and faster progression of colon adenomas with an increased risk of progression in

CRC [20]. It is important to underline that colonic polyposis are classified as advanced if the following conditions are present: diameter above 1 cm, villous form for histopathology analysis. According to the data of scientific literature, high grade dysplasia is found in 25% on surveillance colonoscopy if performed at 1–2-year intervals [27], and the recurrence of neoplastic lesions is more frequent in CF patients with colonic polyposis [39]. The starting screening at age 30 years in CF patients may permit to find polyps and, sometimes, tumors, but it is not confirmed by guidelines, and a balance between the risks of screening and the benefits of early detection is needed [21].

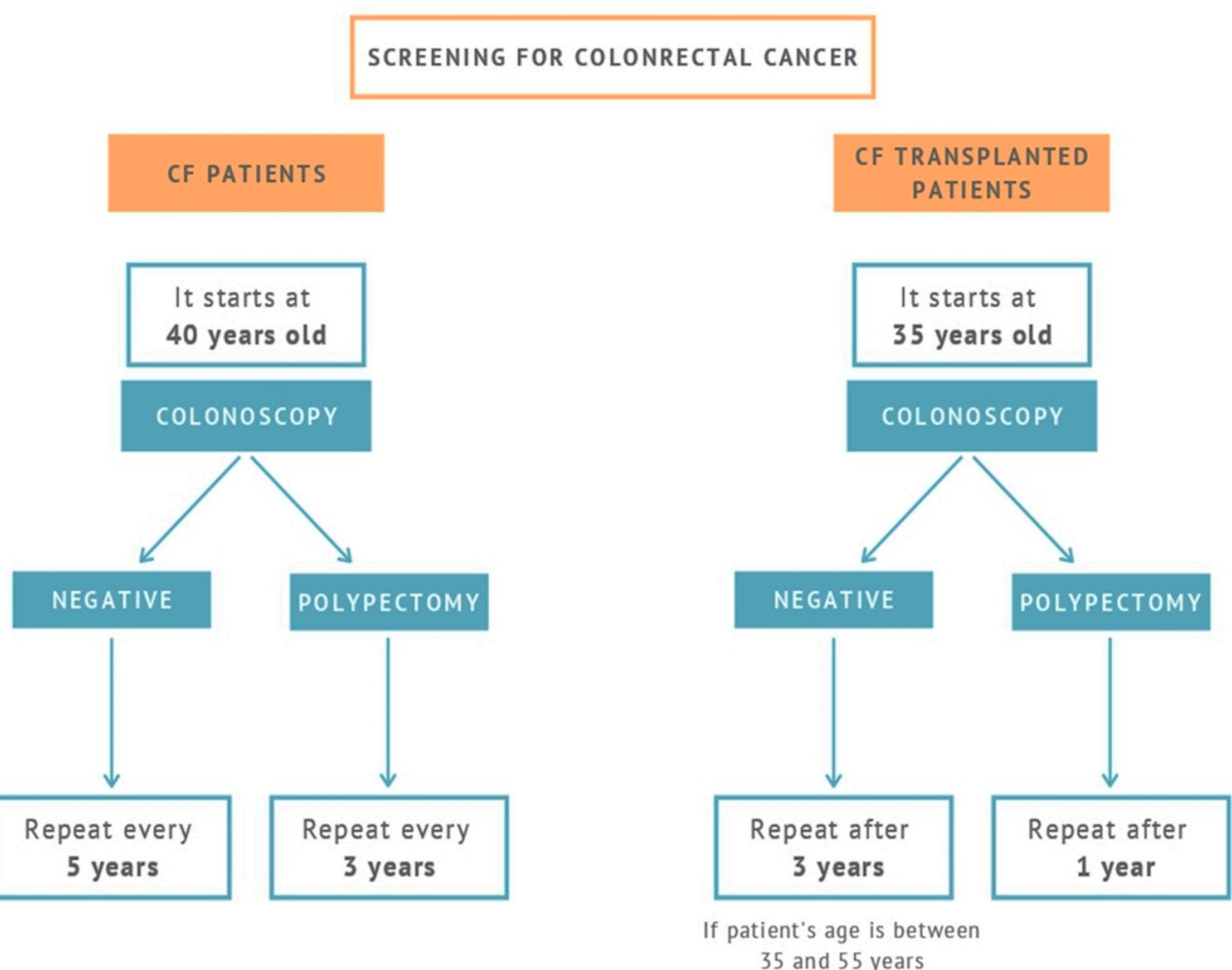

**Figure 2.** Screening for CRC in CF patients.

## 12. Role of Colonoscopy in CF Transplanted Patients

In transplanted CF patients, the Task Force of Cystic Fibrosis suggests colonoscopy screening should start from the age of 35 if the organ transplant occurred at the age of 30 years [21]. In transplanted CF patients before the age of 30, colonoscopy screening should start from the age of 30 [21]. The risk of CRC in transplanted CF patients is greater than in Lynch Syndrome (LS), considering that LS is one of the most common hereditary CRC syndromes [20], and it is always important to consider the presence of very large polyps (>3 cm in diameter) and multiple polyps in the colon (number > 10). The current consensus recommendations about CRC in CF patients are supported by the cost effectiveness analysis. Data for the cost effectiveness analysis were obtained from simulation models for CRC [52]. Abraham et al. consider that the optimal interval for the second colonoscopy screening in CF transplanted patients with a prior negative examination should be considered after 3 years if patients' age is between 35 and 55 years [53]. In case of very large polyps, multiple

polyps or high-grade dysplasia, the surveillance interval should be shortened, and the patient should undergo a new colonoscopy within 1 year [54].

### 13. Role of Bowel Preparation

The management of CRC screening in CF patients should be provided by dedicated endoscopists and, particularly, familiar with the preparation required for bowel cleansing [21]. A correct bowel preparation is essential for an optimal detection of colon polyps [55]. However, bowel preparation can be difficult in CF patients [21]. The main elements for a better bowel preparation include the following advice [2]: (1) several smaller volume washes are better than a single larger volume wash, (2) colonoscopy should be performed as soon as possible after the last wash, (3) the patient should be informed about the importance and the instruction of bowel preparation [12]. For CF patients, it is recommended an increased-intensity regimen regarding the bowel preparation before the programmed colonoscopy [56]. Polyethylene Glycol combined with an electrolyte solution (not sport drinks) is commonly used in the general population [57]. These preparations may have a role in CF patients, but it is important to highlight that if these preparations are administered, they can cause an electrolytic shift in CF patients [58]. A correct bowel preparation allows a better endoscopic quality and colonoscopy measures [52] with a higher precision for the diagnosis of adenomatous lesions. Moreover, endoscopists can perform a correct cecal intubation, within a reasonable inspection time [59]. A previous study has demonstrated that the colonoscopy preparation is more time-consuming in CF patients than in other patients [16]. The risk of complications and adverse events during colonoscopy is low, but CF patients have also an increased risk of these events [60]. Matson et al. [61] suggested to use a bowel preparation modified specifically for CF patients. The modified CF preparation protocol provides the following indications before colonoscopy: (1) an osmotic laxative daily 14 days beforehand; (2) a low-fiber diet 8 days beforehand; (3) 1 sachet of magnesium citrate in 250 mL, 3 Bisacodyl TM tablets and 3 sachets of Glycoprep C TM in 3 L of fluid/day 3 days beforehand; (4) if DIOS history is present, 3 sachets of Glycoprep C TM in 3 L of fluid/day 2 days before, (5) 3 sachets of Glycoprep C TM in 3 L of fluid/day the day beforehand; (6) 1 sachet of Glycoprep C TM in 1 L of clear fluid on the day of the procedure [61]. The efficacy of this bowel preparation was evaluated by Queensland Bowel Cancer Screening Program Bowel Preparation Descriptor Scale [62] modified from the Boston Bowel Preparation Scale (BBPS) [63]; the parameters of this scale are Excellent, Good, Fair or Poor. Matson et al. [61], in their paper, have demonstrated that CF Bowel preparation group had a higher proportion of "excellent/good" colonic cleanse than standard preparation (50% versus 25.9%) and lower rate of "poor" cleanse (10.5% versus 44.5%); "fair" cleanse was similar between the 2 groups (39.4% versus 29.6%). The detection rate of an adenomatous polyp at the first colonoscopy was higher in CF bowel preparation than standard preparation (50% versus 18.5%) [63].

### 14. Future Considerations

In the near future, the recommendation of Cystic Fibrosis Task Force for prevention of CRC [52] will be re-evaluated because the therapy with CFTR modulators could modify the incidence of CRC, based on penetrance condition, on the onset age of the disease and on the type of treatment carried out [53]. Actually, it is not possible to evaluate the different CRC incidences with the use of CFTR modulators because they have been on the market for a short time, not allowing to measure life expectancy. These recommendations are valid both for asymptomatic and symptomatic CF patients (Table 1) [21]. All physicians should consider CF as a "a colon cancer syndrome". The CF patients with a severe genotype have a higher risk of CRC than CF patients with milder mutations [9]. Milder mutations are present in a small number of CF patients (<10%), and these patients have a longer survival, and the onset of CRC is delayed respect to other CF patients [9]. Gory et al. [17], in an Australian screening study, found 4 CRC and 1 ileal tumor in CF patients with Delta F508 Homozygosis mutation. However, the onset of CRC in older CF patients is unclear, and

new studies should be conducted to define the cancer risk in older patients with milder CF mutations [4]. These patients, in the future, will have an increased survival related to the development of a new class of drugs that can repair the mutation of CFTR gene as Lumacaftor, Ivacaftor, Tezacaftor, Elexacaftor (the last 3 drugs are used in "Combo" therapy) and denominated "Modulator Agent" [64]. However, the CFTR mutations (which determine CF) are relatively few compared to the total of the recorded CFTR mutations, which are more than 1900 [63]. In the future, a greater knowledge of the biology and genetics of the CFTR gene and its role in the CRC will be required to allow the use of new therapies and a more integrative approach [65]. The therapy with modulator drugs, as Ivacaftor (IVA), may determine a change in the CF patients Gut Microbiota. Effects reported are a decreasing intestinal inflammation due to an increased number of *Akkermansia muciniphila*, a gram-negative mucin degrading bacterium that reduces intestinal inflammation [66]. IVA may also determine an improved nutritional status, a marked reduction of chronic abdominal symptoms and the development of CRC, but further studies are required to confirm these results [67]. Screening Colonoscopy does not prevent the development of other gastrointestinal neoplasms. CF Foundation recommends that all decisions on CRC screening and surveillance in CF patients are always based on a common decision between provider and individual with CF about treatment, comorbidities, safety and quality of life [11].

**Table 1.** The nine rules to respect regarding colon rectal cancer (CRC) in cystic fibrosis (CF) patients.

| |
|---|
| (1) Colonoscopy should be considered the most important screening examination for CRC screening in CF patients |
| (2) Computed Tomography Colonography (CTC), Blood Occult and DNA stool screening test, sigmoidoscopy are not recommended as screening tests for CF patients |
| (3) CRC screening should be started at age 40 in CF patients and rescreening should continue every 5 years |
| (4) In presence of adenomatous polyps, a colonoscopy should be performed every 3 years and annually in presence of large and multiple polyps |
| (5) In patients with CF of 30 years of age and subjected to solid organ transplantation, colonoscopy is indicated after 2 years from transplantation except in subjects with negative colonoscopy performed in the last 5 years |
| (6) In case of diagnosis of adenomatous polyps in transplanted patients, surveillance colonoscopy should be performed every 3 years and shorter intervals if adenomatous polyps are large (>1 cm in diameter) and multiple (if it is possible, every year, and if negative colonoscopy, every 3 years) |
| (7) CF foundation recommends an intensive regimen for bowel preparation in CF patients with 3–4 washes (minimum of 1 l purgative per wash) with the last wash occurring within 4–6 h before examination |
| (8) CF Foundation recommends that all decisions about CRC screening and CF patient should be managed by CF health care professional and endoscopist |
| (9) CF Foundation recommends that all decisions on CRC screening and surveillance in CF patients be always based on a common decision between provider and individual with CF about treatment, comorbidities, safety and quality of life |

## 15. Conclusions

At this moment, colonoscopy screening should be considered as the best way to guide CF patients management and prevention for CRC and other gastrointestinal neoplasms because it allows to prevent adenoma cancerization or diagnose and treat the CRC earlier [4]. Chemo-preventive agents such as cyclooxygenase inhibitors (e.g., aspirin) and 3 hydroxy 3 methylglutaryl-coenzyme A reductase inhibitors (Statins) could be useful [68] to reduce the risk of developing CRC. The life expectancy of CF patients could increase still more in the next years, and CRC screening will play an important role in that. More CFTR modulator drugs will be available on the marketplace in the future, and they may improve

the natural history of CF patients. The gastroenterologist must be aware of the multiple gastrointestinal manifestations of CF, and the screening of CRC in CF patients represents a stimulating challenge to improve further on the survival of these patients. Our goal is to explain the opportunity to research CRC in CF patients, but further population studies are needed to confirm the necessity to screen this type of patients. Wong et al. [69], in their work, reported that Italy had a decrease in CRC incidence among persons younger than 50 years, and the knowledge of the increased risk of CRC in CF patients could permit to improve this data and to lower the percentage of deaths for CRC.

**Funding:** This research received no external funding.

**Conflicts of Interest:** The authors declare no conflict of interest.

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
