# Peer review of "Surveillance of Colorectal Cancer (CRC) in Cystic Fibrosis (CF) Patients"

_gastrointestdisord, doi:10.3390/gidisord3020009_

Round 1

Reviewer 1 Report

It is a nice overview of the screening possibilities in people with cystic fibrosis. However, I am missing the potential complications that go with colonoscopy. Do the advantages of colonoscopy screening outweigh the disadvantages.

I would also check the English writing and use of certain terms. Also the use of abbreviations when necessary needs work. For example: if you explained an abbreviation once, you can use the abbreviation

Author Response

We correct and add your suggestions

Thank you very much

Giovanni Casella 

Reviewer 2 Report

Overall great topic and very important. But needs substantial rewrite to clarify thoughts.  The bones are there, but needs work.

ABSTRACT

The whole abstract needs to be rewritten.

The sentence about DIGEST seems randomly placed (line 34-46) – there needs to be a connecting sentence about the presence and role of CFTR in the GI tract. Lines 40-44 need to be rewritten. Start with the last sentence and then define when colonoscopy should be done based on age and transplant status.

BODY

Line 57 – “most common” instead of “commonest”

Line 67 – “the functional” instead of “their”

Lines 80-84 – need to bring back something from the 1st paragraph or add some information about how CF is a multisystem disease. The sentences are disjointed. Bring in the GI tract again – then you conclusion in line 85-87 will make more sense.

Line 93-93 should probably be the 2nd sentence of this section on CF and Colon Cancer – after 1st sentence in line 89.

Line 120 – is there a reference for low physical activity? Is that a true statement? Physical activity is promoted in CF.

Line 121 – I think we need to call it CF related diabetes.

Line 134 – Are you referring to one chloride channel or multiple?

Lines 134-140  Appear to be one long run-on sentence. Is there supposed to a period in line 139 are the [34] and then The GI tract?

Line 143 – Add “which is a” before “typical…”

Line 145 - ? “thank” does not make sense

Line 154 – “they” doesn’t make sense – sentence need reconstruction – although appropriate thought process is there

Line 166 – “ii” should be “is”

Line 168 – I don’t think you need the “but” before CRC

Line 189 – change “thank” to “thanks”

Line 190 – sentence does not make sense – again I know what you are trying to say – but not grammatically correct.

Line 195 – Add “The” before “Cystic Fibrosis…”

Line 197 – in some places you say screening in transplanted patients should start at age 30 and other ages 35. I am pretty certain its age 30.

Line 207 - ? “whole CF patients” – grammatically incorrect – could simply say does not recommend and then say “however, there are studies that suggest…”

Line 215 – check grammar of this sentence

Line 230 – check grammar – “The starting screening age”  - are you referring to immunosuppressed or not immunosuppressed CF patient?

Line 254 – take out “is needed an”

Line 255 – “polyethylene glycol combined with an electrolyte solution” might be better than “sport drinks”

Table 1 - #7 – capitalize Foundation

Line 310-14  – These sentences don’t make sense. This is not a strong ending – please consider rewriting.

Author Response

(The authors gave the same response as above.)

Reviewer 3 Report

The authors performed a review in order to provide an overview of colorectal cancer in cystic fibrosis. This includes the epidemiology, pathophysiology and surveillance options. It provides a good overview of what is known.

The study is well written, however, the different subjects are not always well structured. I would help to combine all data about the epidemiology, pathophysiology and surveillance strategies, now this is written down throughout the main text. Furthermore, the aim of the review is not described in the main text. Additionally, a suggestion is to change the nomenclature of screening to surveillance. As cystic fibrosis patients can be considered a high-risk-group of developing colorectal cancer, surveillance is aimed while in an average-risk population screening is performed. Furthermore, aberrations are written out more than once quite often.

Title; Suggestion to alter screening to surveillance.

Abstract;

The fact that this manuscript is a review became clear in the last sentence. Maybe state this sooner. Furthermore, the abstract only focusses on surveillance, however, the main text includes much more information about the pathophysiology as well etc.

Introduction;

Do you know how often the GI-tract is involved in cystic fibrosis?

The aim of this review is not stated in the introduction. Furthermore, why did you choose CRC of the other gastrointestinal diseases which could be involved in cystic fibrosis?

Page 5. Line 88: I would combine this section within the introduction and the end of the epidemiology section. First explain why CRC, and then the general epidemiology.

Pg. 5 line 107: do you know the SIR of CRC in cystic fibrosis patients?

Section pathophysiology:

Suggestion to first explain what CFTR does, what happens if there is a defect and then what it does in the GI-tract.

Pg 5. Line 124: Dysbiosis is mentioned, but another section on page 7 a separate section is stated. Is it possible to combine these to parts or more the section about dysbiosis after this section.

Section screening: This part is somewhat already mentioned earlier in the main text. Furthermore, this does not involve screening but more about the risk of developing CRC in this population. I would suggest to move it more to the top to make an statement why surveillance should be offered to these high-risk individuals for developing CRC.

Pg 7 line 169: is the mortality of CF patients increasing over time?

Pg 7 line 180: What is known about these microbiome changes in average-risk populations for developing CRC?

Section CF patients are at high risk for CRC;

I do find this text more about the epidemiology, and somewhat copies text previously mentioned. Try to make a section about CF in general and there life-expectancy, the risk of developing CRC/adenomas.

Section CFTR carriers;

Move this section also more to the part about epidemiology. Do you know the risk for developing CRC in specifically CFTR carriers? What is the relative risk? Can they also be considered a high-risk group (relative risk ≥ 2.5 compared to an average risk population)? What surveillance strategy would they benefit from? Why mention the average age for CRC and gastric cancer in the general population?

Section CRC surveillance;

I would start with the data of colonoscopy, than explain why not to choose for the other surveillance strategies. For the section about FIT, you also mention multi-target stool DNA test, so maybe alter nomenclature of section to stool tests.

Pg 8. Line 227: Do you know what the characteristics were of the adenomas detected in CF patients? Did dysplasia occur also more frequently in that specific population?

Pg. 8 Line 229: Why the age of 30 years? The a priori chance of developing CRC at that age is very low. At what age were the adenomas detected in the CF patients? Do you recommend another starting age than the Cystic Fibrosis Task Force?

At which age do you recommend to stop surveillance, taking into account there life expectancy?

Pg 8. Line 219: specify why CTC is not recommended?

Pg. 8 line 212: do you know anything about the participation rate of colonoscopy in CF patients?

Section Role of colonoscopy in CF transplanted patients;

Pg 9. Line 244: classify very large polyps and multiple polyps.

Section Future considerations;

Pg 10. Line 278: I did not recall that you mentioned that the CRC incidence was different between the different CFTR modulators, or that the treatment is changing. Maybe good to add a section about this in the part of life-expectancy.

Pg. 10 line 282: does CF mainly occur in the colon or also in the rectum?

Do you think there is a role for stool test surveillance?

Section conclusion;

Pg 11. Line 302: you do not prevent the development of other gastro-intestinal neoplasms by performing a colonoscopy.

Pg. 11 line 303: These chemo-preventive agents are not mentioned in the main text.

Table 1. Point 9 was not mentioned in the main text.

Figures are too dark but the content is clear.

Author Response

(The authors gave the same response as above.)
